# Interactions between West Nile Virus and the Microbiota of *Culex pipiens* Vectors: A Literature Review

**DOI:** 10.3390/pathogens12111287

**Published:** 2023-10-27

**Authors:** Marta Garrigós, Mario Garrido, Guillermo Panisse, Jesús Veiga, Josué Martínez-de la Puente

**Affiliations:** 1Department of Parasitology, University of Granada, 18071 Granada, Spain; m.garrido@ugr.es (M.G.); jveiga@ugr.es (J.V.); jmp@ugr.es (J.M.-d.l.P.); 2CEPAVE—Centro de Estudios Parasitológicos y de Vectores CONICET-UNLP, La Plata 1900, Argentina; gpanisse@cepave.edu.org; 3CIBER de Epidemiología y Salud Pública (CIBERESP), 28029 Madrid, Spain

**Keywords:** flavivirus, insect vectors, mosquito-borne pathogens, mosquito microbiota, vector competence, *Wolbachia*

## Abstract

The flavivirus West Nile virus (WNV) naturally circulates between mosquitoes and birds, potentially affecting humans and horses. Different species of mosquitoes play a role as vectors of WNV, with those of the *Culex pipiens* complex being particularly crucial for its circulation. Different biotic and abiotic factors determine the capacity of mosquitoes for pathogen transmission, with the mosquito gut microbiota being recognized as an important one. Here, we review the published studies on the interactions between the microbiota of the *Culex pipiens* complex and WNV infections in mosquitoes. Most articles published so far studied the interactions between bacteria of the genus *Wolbachia* and WNV infections, obtaining variable results regarding the directionality of this relationship. In contrast, only a few studies investigate the role of the whole microbiome or other bacterial taxa in WNV infections. These studies suggest that bacteria of the genera *Serratia* and *Enterobacter* may enhance WNV development. Thus, due to the relevance of WNV in human and animal health and the important role of mosquitoes of the *Cx. pipiens* complex in its transmission, more research is needed to unravel the role of mosquito microbiota and those factors affecting this microbiota on pathogen epidemiology. In this respect, we finally propose future lines of research lines on this topic.

## 1. Introduction

Mosquitoes are considered a major concern for public health, wildlife and livestock, as vectors of numerous pathogens such as haemosporidians of the genus *Plasmodium*, nematode worms causing lymphatic filariasis, and a number of arboviruses including Dengue virus, Zika virus and West Nile virus, among others [1,2,3,4]. Mosquito vectorial capacity, which provides information on the epidemiological relevance of a vector in the transmission of a given pathogen, depends on different factors including the mosquito survival rate, pathogen extrinsic incubation period and vector competence [5]. Environmental conditions are known to affect all of these components of the vectorial capacity including abiotic factors such as temperature and humidity, and biotic factors such as predation risk and competition of mosquito larvae [5,6]. In addition, intrinsic factors of mosquito vectors affect the components of vectorial capacity, with the mosquito gut microbiota composition being one of them. Mosquito gut microbiota could affect either positively or negatively the pathogen transmission through its effects on pathogen susceptibility, pathogen development, vector density, vector survival and vector behavior (see [5] for a recent review).

## 2. Relevance of West Nile Virus

The West Nile virus (WNV) is a Flavivirus (family Flaviviridae) that naturally circulates between mosquitoes and birds. In addition, WNV occasionally infects other vertebrates [7], including humans, causing a broad range of clinical symptoms: from asymptomatic or a mild febrile illness to myocarditis and encephalitis, occasionally causing the death of infected individuals [8]. Avian hosts contribute to the virus circulation producing a level of viremia allowing the WNV transmission to mosquitoes while humans and other mammals act as ‘dead-end’ hosts [9]. West Nile virus is biologically diverse, with lineages 1 and 2 being the most widespread and epidemiologically relevant to birds, humans and horses [10,11,12].

The WNV was discovered in 1937 in Africa (Uganda) [13], and subsequently identified in Europe (France) [14], Asia (Iran) [15] and Oceania (Australia) [16], with a number of outbreaks and isolated cases reported in these continents since then [17]. For example, during the last few decades, WNV outbreaks were reported in Italy, Greece, Russia, Israel and Turkey, among other countries [18]. Particularly relevant was the 2018 outbreak in Europe where 2083 people were infected by the virus, with 181 people dying in the continent [19]. More recently, different outbreaks have also occurred in other Mediterranean countries. In Spain, although the circulation of WNV was repeatedly reported in birds and horses during the last decades [20,21], human cases have been reported sporadically with the largest known outbreak being recorded in 2020, resulting in 77 people infected (40 confirmed and 37 probable) and 7 fatalities [22].

The WNV acquired special relevance after its introduction in North America, where caused the largest epidemics of neuroinvasive WNV disease in humans ever reported. There, WNV was identified for the first time in New York in 1999 [23] and, subsequently, the virus spread rapidly across the United States, to the South across Mexico [24] and the Caribbean [25], and to the North in Canada [26]. Finally, the virus has also been registered in South America [27,28]. Up to date, at least 2773 human fatalities due to WNV disease have occurred in the US [29]. Additionally, WNV caused a large ecological impact on avian populations, with 47,923 dead birds belonging to 294 species from 1999 to 2004 [30], with a large impact on some species that did not recover since pathogen introduction [31]. For instance, the American crow (*Corvus brachyrhynchos*) population declined by up to 45% since the WNV introduction in this area [32].

## 3. The Role of *Culex pipiens* Complex in WNV Transmission

West Nile virus is transmitted by mosquito species belonging to several genera [33] with different species playing a predominant role in its transmission in different areas [34]. Among them, species of the genus *Culex*, particularly those of the *Culex pipiens* complex, play a predominant role in WNV epidemiology [35]. In addition to their role in the transmission of other pathogens including zoonotic ones [35], mosquito females of the *Cx. pipiens* complex are considered crucial for the WNV circulation between birds, but also for its transmission to humans and horses [36].

The *Cx. pipiens* complex comprises *Cx. pipiens* pipiens—including its two forms or biotypes, pipiens and molestus, and their hybrids—*Culex pipiens* pallens, *Culex quinquefasciatus*, *Culex australicus* and *Culex globocoxitus* [35,37]. Among them, *Cx. pipiens* and *Cx. quinquefasciatus* stand out as vectors of WNV, where the presence of this pathogen has been frequently recorded in mosquitoes captured in the wild [33,38,39]. Specifically, *Cx. pipiens* is a major vector of WNV in Europe and Africa [40,41], being also a competent vector in North America and some areas of South America together with *Cx. quinquefasciatus* [38,42,43,44]. Furthermore, *Cx. quinquefasciatus* is also considered a WNV vector in Australia and Asia [45,46]. These species are particularly relevant for the transmission of WNV due to, among other factors, their wide distribution range through all continents except Antarctica [35], their high vector competence for this virus [47,48] and their ability to feed on both birds and mammals, including humans [35,49,50].

## 4. Mosquito Microbiota and Pathogen Transmission

Except for the obligate intracellular symbionts that are predominantly maternally transmitted, mosquitoes acquire their gut microbiota during the larval stage from their breeding waters [51,52,53]. During the metamorphosis, most taxa of the larval gut microbiota are expelled in a meconium [54], while a small proportion of this microbiota reach the adult stage through transstadial transmission [53,55]. Thus, the composition of the microbiota of adult mosquitoes is largely determined by the larval environment, including spatial and temporal factors such as the presence of pollutants in the breeding water, and can vary between individuals and species [56,57]. Thereafter, adult microbiota is modulated by their feeding sources, including plant sugars, and, in the case of female mosquitoes, by their blood meal sources [56,58].

There is growing evidence supporting the role of mosquito gut microbiota as a major driver of the responses of mosquitoes against the pathogens interacting with them, finally affecting different components of the mosquito vectorial capacity [5,59]. Among others, mosquito microbiota affects the survival rate of mosquitoes [60,61], their vector competence [62,63] and the pathogen extrinsic incubation period [64,65]. In addition, mosquito microbiota may affect the development of pathogens in mosquitoes through direct and indirect effects including the competition with the pathogens for resources [66], the hindrance of necessary interactions between the pathogen and vector epithelium [67], the secretion of anti-pathogen molecules [68], the formation of the peritrophic matrix around the blood bolus after blood feeding, which is a barrier against pathogens [69], or the activation of immunological responses [66,70,71].

The effects of mosquito microbiota in pathogen infections depend on the mosquito species, pathogen strain, and the symbiont taxa studied [71]. Although mosquito microbiota is composed of bacteria, fungi and protozoan microorganisms [72], bacteria are the most studied component of the mosquito microbiota. In particular, the intracellular symbionts of the genus *Wolbachia* have been extensively demonstrated to affect the reproductive phenotype of mosquitoes [73] and their resistance to virus and protozoan infections, both enhancing [74] and blocking the infection [66,75,76]. Other symbionts including bacteria of the genera *Chromobacterium*, *Proteus* and *Paenibacillus* have been identified as potential factors inhibiting viral infections [77,78], while *Serratia* spp. has been reported to enhance them [79,80].

On the other hand, pathogen infection may also shape mosquito microbiome, including microbial load and microbiota composition. For example, *Plasmodium* parasites have been observed to reduce bacterial load in *Anopheles stephensi* mosquitoes [81], and arboviral infections may alter the microbial community of mosquitoes belonging to the *Aedes* genus [82,83].

## 5. Interactions between WNV and the Microbiota of Mosquitoes of the *Culex pipiens* Complex

Here, we review the published studies on the interactions between the microbiota of mosquitoes of the *Culex pipiens* complex and WNV. To date, the authors have investigated the relationships between one (a single bacterial genus, e.g., *Wolbachia*) or more components of the microbiota and WNV infections in mosquitoes of the *Cx. pipiens* complex, including *Cx. pipiens* and *Cx. quinquefasciatus.* This information is summarized in Table 1.

### 5.1. Studies on Culex pipiens Mosquitoes

Novakova et al. [57] and Leggewie et al. [84] performed correlative studies on the interaction between WNV infection and mosquito microbiota composition and *Wolbachia* infections, respectively. Novakova et al. [57] used *Cx. pipiens*/*Cx. restuans* mosquitoes, and took into consideration the spatial (sampling region) and temporal (different seasons over 3 years) variability, including climatic variables. They found that seasonal shifts in microbiota were associated with patterns of WNV prevalence, with higher temperatures correlating with lower relative abundance of *Wolbachia* and higher WNV prevalence in mosquitoes. Moreover, the relative abundance of *Wolbachia* was significantly higher in WNV negative mosquitoes compared to those WNV positive. In contrast, using *Cx. pipiens* mosquitoes, Leggewie et al. [84] found no significant differences in *Wolbachia* load between WNV-positive and WNV-negative mosquitoes, although a positive correlation between *Wolbachia* and WNV load in infected mosquitoes was found.

Experimental approaches have also been conducted to test the potential association between *Cx. pipiens* microbiota and WNV development. Zink et al. [85] fed wild female mosquitoes with non-infectious blood meals or blood meals containing WNV (NY1986 strain). Seven days later, the authors tested for the WNV infection of exposed mosquitoes and compared the bacterial richness and load of different groups among the unexposed and exposed mosquitoes, including those negative and positive mosquitoes after WNV exposure. A higher bacterial diversity was associated with WNV exposure and even higher when mosquitoes were infected with WNV. In concordance with Novakova et al. [57], the mean relative abundance of bacteria of the genus *Wolbachia* in WNV-infected mosquitoes was significantly lower than in WNV uninfected and unexposed ones. Furthermore, exposed and WNV-infected mosquitoes showed a higher relative abundance of bacteria of the genera *Enterobacter* and *Serratia* than unexposed ones, suggesting that bacteria of these genera could play a role in WNV development in mosquitoes.

Micieli & Glaser [86] further assessed the interaction between *Wolbachia* load and WNV infection using *Cx. pipiens* mosquitoes from a colony naturally infected with *Wolbachia*. In this study, authors fed mosquitoes with a WNV (WNV02) infected blood meal to obtain mosquitoes with three different degrees of virus development: (i) non-disseminated infections (uninfected mosquitoes); (ii) mosquitoes with a disseminated infection, that is those positive for the presence of WNV in their legs; and (iii) transmitting mosquitoes, those with positive results for the presence of WNV in their legs and saliva. The authors did not find any significant correlation of *Wolbachia* somatic densities with the WNV infection status of mosquitoes, which contrasts with results reported by Zink et al. [85].

### 5.2. Studies on Culex quinquefasciatus Mosquitoes

Alomar et al. [87] and Glaser & Meola [88] used a similar experimental design, starting from *Cx. quinquefasciatus* colonies naturally infected by *Wolbachia* to finally obtain an uninfected line using the antibiotic tetracycline. They fed mosquito females with blood containing WNV, and then compared the rates of viral body infection, dissemination and transmission, and the WNV load between *Wolbachia*-infected and uninfected individuals at 14 days post feeding [87] and at 5, 7 and 14 days post feeding [88], respectively. Alomar et al. [87], using the *Wolbachia* strain *w*Pip, also considered in their analyses the competition during the larval stage of mosquitoes. These authors found that *Wolbachia* infection significantly reduced the WNV load of infected mosquitoes only when larvae were exposed to low-competition stress treatment. No significant differences were found in the WNV body infection, dissemination or transmission rates according to the *Wolbachia* infection status. Consistent with this, Glaser & Meola [88] found a significantly higher WNV load in *Wolbachia* uninfected mosquitoes. In addition, WNV dissemination and transmission rates were significantly higher in *Wolbachia* uninfected mosquitoes at all time points. On the other hand, Shi et al. [89] compared the bacterial diversity of wild-collected *Cx. quinquefasciatus* mosquitoes exposed to three treatments: (i) 10% sucrose solution diet; (ii) noninfectious blood meal; (iii) bloodmeal containing WNV (PaAn001 strain). However, no significant differences in bacterial diversity after WNV exposure were found between treatments at 7 and 14 days post feeding.

### 5.3. Additional Relevant Studies on Species Other Than the Culex pipiens Complex

In addition to the studies on mosquitoes of the *Cx. pipiens* complex, the role of mosquito microbiota in the development of WNV has also been investigated in other species of mosquitoes, including those of the *Culex* genus. These studies have been mainly focused on addressing the role of *Wolbachia* in the development of WNV in these insects. This is the case of the study by Dodson et al. [90], who experimentally infected *Culex tarsalis* mosquitoes from a colony with *Wolbachia* (*w*AlbB strain) and/or WNV. These authors compared the WNV infection, dissemination and transmission rates according to *Wolbachia* infection status and found that WNV infection was significantly higher in *Wolbachia*-infected than in *Wolbachia*-uninfected mosquitoes. In addition, using a cell line of *Aedes aegypti* mosquitoes, Hussain et al. [91] tested the effect of *Wolbachia* infection on WNV replication. These authors found a higher accumulation of WNV RNA in *Wolbachia*-infected cells.

Discrepancies between studies and species of mosquitoes further support the necessity to identify the role of mosquito microbiota in the transmission of WNV. This is especially relevant because although the microbiota of mosquitoes is largely determined by the environmental conditions of the breeding sites, clear differences exist between mosquitoes of different species but of the same origin [84,92], which together with other ecological factors such as host preferences, may modulate the transmission of vector-borne pathogens. This may be especially relevant for the case of other species of the *Culex* genus, which along with *Cx. pipiens*, may play a key role in the local circulation of WNV in different areas [93,94].

### 5.4. Potential Factors Explaining the Interactions between Mosquito Microbiota and WNV

Novakova et al. [57] proposed that the negative correlation they observed between *Wolbachia* abundance and WNV infection could be explained by the widely reported *Wolbachia* immuno-modulatory capacity in mosquitoes [51], which has also received experimental support [86,87]. On the other hand, Zink et al. [85] proposed the alternative is WNV infection which reduces *Wolbachia* relative abundance, either by direct inhibition or through the immune response modulation. Both hypotheses are not mutually exclusive, as the interaction between mosquito microbiota and pathogens is bidirectional. Mosquito innate immune pathways could be shared in the response against bacteria, fungi, protozoans and viruses, such as the Toll and the immunodeficiency (IMD) pathways [95]. For example, Zink et al. [85] showed that the infection with WNV increased the bacterial diversity of *Cx. pipiens* and was associated with an up-regulation of classical invertebrate immune pathways. Furthermore, it has been reported that *Wolbachia* infections alter the profiles of several mosquito miniRNAs that are involved in antiviral responses [96], including those against WNV [97]. Therefore, the experimental infection by *Wolbachia* may alter the mosquito antiviral response, and vice versa. Finally, it is important to consider whether *Wolbachia* inhibits or enhances viral infections depends on virus identities and *Wolbachia* strains [98,99,100], the mosquito host species and the nature of the *Wolbachia*–host interaction [75,101].

## 6. Conclusions and Future Directions

The studies published so far evidence that the vector competence of mosquitoes of the *Cx. pipiens* complex for WNV is modulated by *Wolbachia* infection in either way. While correlative studies in *Cx. pipiens* showed no clear patterns in the relationship between *Wolbachia* and WNV, experimental studies in *Cx. pipiens* and *Cx. quinquefasciatus* support that there is a negative correlation between *Wolbachia* and WNV infections (see a summary of these studies in Table 1). The important role of this endosymbiont may potentially explain differences between mosquito species and populations in the transmission of WNV, as differences in *Wolbachia* load exist [102,103]. In addition, most of the articles reviewed here do not specify the *Wolbachia* strain studied, which could also affect the interactions between mosquitoes and WNV.

Although most studies were conducted on *Wolbachia*, other bacterial genera may be also relevant in the interaction between mosquito microbiota and mosquito-borne pathogens, showing different effects. *Enterobacter* and *Serratia* are found in higher proportions in WNV-exposed *Cx. pipiens* mosquitoes, including infected and uninfected ones [57]. Previous studies on *Ae. aegypti* suggest that *Serratia* spp. may facilitate Dengue [79,104] and Chikungunya [80] viral infections, by suppressing the immune response of mosquitoes by the secretion of (1) a polypeptide that interacts with a mosquito protein required for virus infection in mosquitoes [105], and (2) a protein that digests membrane-bound mucins on the mosquito gut epithelia allowing virus dissemination [104]. Similar processes could affect *Cx. pipiens*–WNV interactions, potentially affecting the susceptibility of mosquitoes to viral infections. Thus, the presence of other bacterial taxa affecting WNV transmission and potentially masking the effects of *Wolbachia* should be considered.

Moreover, different bacterial taxa of the mosquito microbiota can interact with each other, with potential implications for pathogen transmission [106]. For example, *Wolbachia*-infected *Ae. aegypti* mosquitoes showed a reduced relative abundance of a large proportion of bacterial taxa compared to *Wolbachia*-uninfected mosquitoes [107]. In addition, *Chromobacterium* showed antibacterial activity against many bacterial species commonly found in the midgut of *Aedes* and *Anopheles* mosquitoes [78]. Hence, studies including more than one bacterial group, ideally the whole microbiome, and their interactions, are essential to understand how the mosquito microbiota affects WNV transmission in the wild. These studies should also consider the potential role of antimicrobial environmental pollutants on mosquito microbiota under natural conditions. Pharmacological pollutants such as antibiotics are commonly found in freshwater, potentially affecting the mosquito microbiota during the larval stages. Recent studies have found links between antibiotic driven disruptions of mosquito microbiota and the development of mosquito-borne pathogens [61,108], a pattern that could also affect the development of WNV [51] that merits further studies in the future.

Finally, very few studies have explored the potential role of the microbiota of vertebrate hosts in WNV infections, which could be also a factor affecting the epidemiology of the virus under natural conditions. Host microbiota may modulate the susceptibility of vertebrates to mosquito attacks, which may be affected by mosquito-borne pathogen infections [109]. In addition, recent evidence suggests that bird microbiota could potentially affect the development of pathogens, including viral infections. For instance, in a recent study using influenza, the authors reported that birds treated with antibiotics as a microbiota disruptor increased their susceptibility to influenza infections [110]. In the case of the mosquito-borne avian *Plasmodium*, the infection has been shown to partly shape bird gut microbial community [111]. Furthermore, several cloacal bacterial symbionts have been linked to the survival of *Plasmodium*-infected individuals in a Hawaiian bird population [112]. For the case of WNV, Vaz et al. [113] identified components of their microbiota by bacterial cultures and biochemical tests from cloacal and oropharyngeal samples from red-tailed amazon parrot (*Amazona brasiliensis*) nestlings and screened the infection by WNV and other pathogens. The authors identified 17 bacterial species as components of the nestling’ microbiota but, unfortunately, all samples tested negative for WNV, and therefore the relationship between the virus infection and the composition of the microbiota could not be properly assessed.

## Figures and Tables

**Table 1 pathogens-12-01287-t001:** Summary of the results obtained in correlative (C) and experimental (E) studies on the interaction between WNV infections and the microbiota of mosquitoes of the *Cx. pipiens* complex. See further details in the main text.

Mosquito Species	Type	Symbiont(s)	Microbiota Variable	WNV Variable	Effect	Ref
*Cx. pipiens*	C	*Wolbachia*	Relative abundance	Prevalence patterns, infection	Negative	[57] *
	C	*Wolbachia*	Load	Load	Positive	[84]
	E	*Wolbachia*	Relative abundance	Infection	Negative	[85]
		*Enterobacter Serratia*			Positive	
		Microbiome	Bacterial diversity		Positive	
	E	*Wolbachia*	Load	Infection, dissemination, transmission	None	[86]
*Cx. quinquefasciatus*	E	*Wolbachia*	Infection status	Load	Negative	[87] **
	E	*Wolbachia*	Infection status	Dissemination, transmission, load	Negative	[88]
	E	Microbiome	Bacterial diversity	Infection	None	[89]

* Authors did not differentiate between *Cx. pipiens* and *Cx. restuans* mosquitoes. ** The negative effect was only found in mosquitoes of the low larvae competition treatment.

## Data Availability

No new data were created or analyzed in this study. Data sharing is not applicable to this article.

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
