# Peer review of "Interactions between West Nile Virus and the Microbiota of *Culex pipiens* Vectors: A Literature Review"

_pathogens, 2023, doi:10.3390/pathogens12111287_

Round 1
Reviewer 1 Report
Comments and Suggestions for Authors
Overall a good review on an important subject. However, judging from the review, existing literature is not sufficient to give straightforward answers on the subject.
Some minor comments:
1. The figure (or graphical abstract) is not very informative. The figure attempts to view the interactions between the mosquito, the virus, and larval microbiota. These interactions are not clearly presented. I'm also missing a table with a summary of the studies including, mosquito species, bacterial strands and their effect on the virus.
2. Line 63 (and other places) italics.
3. Line 77: Are you referring to gut microbiota or surface microbiota as well? This is the place to make the distinction.
4. Line 101: Culex pipiend and C. restuans are very different at the larval stage.
Author Response
Comments and Suggestions for Authors
Reviewer 1
Overall a good review on an important subject. However, judging from the review, existing literature is not sufficient to give straightforward answers on the subject.
Some minor comments:
- The figure (or graphical abstract) is not very informative. The figure attempts to view the interactions between the mosquito, the virus, and larval microbiota. These interactions are not clearly presented. I'm also missing a table with a summary of the studies including, mosquito species, bacterial strands and their effect on the virus.
ANSWER: Following your suggestion we have removed the figure from the manuscript. This information has been included in a new table where we summarize the results of the studies on the interaction between WNV infection and the microbiota of mosquitoes from the Cx. pipiens complex. In the manuscript we have included the information you mention when it was provided in the cited reference, as well as other relevant data of these studies (see Table 1 and main text).
- Line 63 (and other places) italics.
A: It appears to be a typographical error when converting the submitted Word file to pdf. We apologize for that. We have carefully revised the main text to avoid this error.
- Line 77: Are you referring to gut microbiota or surface microbiota as well? This is the place to make the distinction.
A: We thank your appreciation. It has been specified that we refer to gut microbiota in several sentences (e.g. lines 20 and 119)
- Line 101: Culex pipiend and C. restuans are very different at the larval stage.
A: We apologize for the misunderstanding. We refer to the fact that authors of ref. [57] did not differentiate between these two species in their study, as mosquitoes were collected as adults. We moved this information to the new table (please, see Table 1)
Reviewer 2 Report
Comments and Suggestions for Authors
This manuscript discusses the potential link between the gut bacteria of Culex mosquitoes and the transmission of the West Nile virus. However, the author provides a limited and superficial introduction to this aspect (this part is only discussed in two short paragraphs in the main text, and the discussion is also shallow).
The main text only presents the results suggesting that bacteria may promote or inhibit the virus, without elaborating on the reasons for these phenomena or the possible mechanisms involved. The manuscript lacks length and substance.
The author should first present all the existing reports on mosquito gut bacteria and then explore several possible phenomena, mechanisms, and potential for disease control related to the interaction with West Nile virus.
Additionally, the current research progress does not offer any compelling conclusions. It is acknowledged that there may be a scarcity of reports on this topic, making it challenging to provide a comprehensive review, based on the current content and the author's presentation.
Author Response
Comments and Suggestions for Authors
Reviewer 2
This manuscript discusses the potential link between the gut bacteria of Culex mosquitoes and the transmission of the West Nile virus. However, the author provides a limited and superficial introduction to this aspect (this part is only discussed in two short paragraphs in the main text, and the discussion is also shallow).
The main text only presents the results suggesting that bacteria may promote or inhibit the virus, without elaborating on the reasons for these phenomena or the possible mechanisms involved. The manuscript lacks length and substance.
The author should first present all the existing reports on mosquito gut bacteria and then explore several possible phenomena, mechanisms, and potential for disease control related to the interaction with West Nile virus.
A: Thanks for your suggestion. In the current version of the manuscript we have extended the Introduction section and provided additional details in Conclusions and future directions and Mosquito microbiota and pathogen transmission subsection. We have included information regarding the different mechanisms explaining the effects of the microbiota on pathogen development. Furthermore, in the Mosquito microbiota and pathogen transmission subsection and in a new subsection called Potential factors explaining the interactions between mosquito microbiota and WNV we discussed the mechanisms involved in the pathogen transmission by mosquitoes according to their microbiota and the potential mechanisms underlying the results obtained by the cited works. In our opinion, these changes have significantly improved the current version of the manuscript, so we are very grateful for this comment. We are completely open to include further details if necessary.
Additionally, the current research progress does not offer any compelling conclusions. While it is acknowledged that there may be a scarcity of reports on this topic, making it challenging to provide a comprehensive review, based on the current content and the author's presentation, it is my belief that this manuscript is not suitable for acceptance and publication.
A: We have provided further conclusions based on current knowledge and highlight the future research lines which are necessary. As reported by other reviewers, this article provides an overview of the state of the art on this topic, which is necessary to summarize and discuss the results obtained in this hot topic.
Reviewer 3 Report
Comments and Suggestions for Authors
In this manuscript, Dr. Garrigos and colleagues review the literature related to the microbiota-mosquito-West Nile virus triad. The authors focus on mosquitoes of the Culex pipiens complex and mostly center the review on the bacteria Wolbachia. Authors highlight that correlational studies showed contradictory results, but experimental results displayed a negative relationship between Wolbachia and West Nile virus.
The topic of this paper is exciting and deserves a review paper regarding it. However, the manuscript needs some improvement before publication.
Please find below suggestions in the hope of improving the manuscript.
Title: Needs to be changed. The manuscript briefly discusses the microbiota concerning WNV and mosquitoes; most of the work cited is regarding Wolbachia. The title should reflect that.
The introduction does not lead to an introduction to the paper but a description of the West Nile virus. A brief introduction followed by a subtopic regarding the WNV would be more appropriate.
The subtitle "Mosquito vectors: the role of Culex pipiens complex" is broad when the only importance of this mosquito complex described is related to the WNV. This section felt superficial, considering Culex mosquitoes' medical and veterinary importance. Expand this section or change the subtitle.
The same critique can be applied to the next section, "Mosquito microbiota". The subtitle is also broad, and the text feels superficial. One suggestion is to include how the microbiota is acquired and how it changes during the mosquito life cycle. However, much more can be added, including references to great reviews that can direct the readers if more information is needed if the authors do not want to dive too deep into the subject.
The section "Interactions between WNV and the microbiota of mosquitoes of the Culex pipiens complex" is confusing. Many studies are described in this section, but thoughts or grouping of the information need to be provided. I found myself re-reading the section several times to come to an understanding that it should have been provided in the manuscript.
In the "Conclusions and future directions" section, the authors discuss the correlative and experimental studies, the absence of patterns regarding some studies, and the negative correlation in others. Why is that? I would like to know the author's thoughts on that, and I suggest adding this information to the text.
The figure 1 needs some work. It is hard to interpret and not well discussed in the text.
Lines 63-73: Italicize scientific names.
Line 97: typo on quinquefasciatus
Line 189: Which mosquito are the authors discussing regarding Dengue and Chromobacterium? I suggest rephrasing it.
Author Response
Comments and Suggestions for Authors
Reviewer 3
In this manuscript, Dr. Garrigos and colleagues review the literature related to the microbiota-mosquito-West Nile virus triad. The authors focus on mosquitoes of the Culex pipiens complex and mostly center the review on the bacteria Wolbachia. Authors highlight that correlational studies showed contradictory results, but experimental results displayed a negative relationship between Wolbachia and West Nile virus.
The topic of this paper is exciting and deserves a review paper regarding it. However, the manuscript needs some improvement before publication.
Please find below suggestions in the hope of improving the manuscript.
Title: Needs to be changed. The manuscript briefly discusses the microbiota concerning WNV and mosquitoes; most of the work cited is regarding Wolbachia. The title should reflect that.
ANSWER: Following your suggestion we have highlighted the role of Wolbachia in the abstract and the keywords, however we have decided to maintain the title because the research question addressed here focus on the role of mosquito microbiota and, not only, in the Wolbachia component (please, see lines 22-23, 24-25).
The introduction does not lead to an introduction to the paper but a description of the West Nile virus. A brief introduction followed by a subtopic regarding the WNV would be more appropriate.
A: Following your suggestion, we have added an introduction preceding the other sections of the manuscript (please, see lines 36-51)
The subtitle "Mosquito vectors: the role of Culex pipiens complex" is broad when the only importance of this mosquito complex described is related to the WNV. This section felt superficial, considering Culex mosquitoes' medical and veterinary importance. Expand this section or change the subtitle.
A: Since this review focuses on WNV, in that section we aimed to expose the relevance of the Cx. pipiens complex in its transmission. According to the reviewer´s comment, we have modified the subtitle to clarify that (please, see line 87).
The same critique can be applied to the next section, "Mosquito microbiota". The subtitle is also broad, and the text feels superficial. One suggestion is to include how the microbiota is acquired and how it changes during the mosquito life cycle. However, much more can be added, including references to great reviews that can direct the readers if more information is needed if the authors do not want to dive too deep into the subject.
A: Following your recommendation we have included a number of modifications in the main text. First, the subtitle has been reformulated as Mosquito microbiota and pathogen transmission. In addition, we have added an opening paragraph that includes the main drivers of the acquisition and modification of the mosquito gut microbiota during the life cycle (lines 109-118), and we expanded some ideas, such as the physiological mechanisms underlying the role of the microbiota in the vectorial capacity (lines 127-128), the directionality of the effect of the mosquito microbiota on pathogen transmission, and the effect of virus infection on the mosquito microbiota (lines 136-144). We have cited several reviews on this topic that may help readers to expand on some of the points, including Strand (2018), Grabielli et al. (2021), Hegde et al. (2015), Shi et al. (2023) and Gao et al. (2020). We are completely open to consider additional relevant literature if necessary.
Gabrieli, P.; Caccia, S.; Varotto-Boccazzi, I.; Arnoldi, I.; Barbieri, G.; Comandatore, F.; Epis, S. Mosquito Trilogy: Microbiota, Immunity and Pathogens, and Their Implications for the Control of Disease Transmission. Front. Microbiol. 2021, 12, 630438, doi:10.3389/fmicb.2021.630438.
Gao, H.; Cui, C.; Wang, L.; Jacobs-Lorena, M.; Wang, S. Mosquito Microbiota and Implications for Disease Control. Trends Parasitol. 2020, 36, 98–111, doi:10.1016/j.pt.2019.12.001.
Hegde, S.; Rasgon, J.L.; Hughes, G.L. The Microbiome Modulates Arbovirus Transmission in Mosquitoes. Curr. Opin. Virol. 2015, 15, 97–102, doi:10.1016/j.coviro.2015.08.011.
Shi, C.; Beller, L.; Wang, L.; Rosales Rosas, A.; De Coninck, L.; Héry, L.; Mousson, L.; Pagès, N.; Raes, J.; Delang, L.; et al. Bidirectional Interactions between Arboviruses and the Bacterial and Viral Microbiota in Aedes Aegypti and Culex Quinquefasciatus. mBio 2022, 13, e01021-22, doi:10.1128/mbio.01021-22.
The section "Interactions between WNV and the microbiota of mosquitoes of the Culex pipiens complex" is confusing. Many studies are described in this section, but thoughts or grouping of the information need to be provided. I found myself re-reading the section several times to come to an understanding that it should have been provided in the manuscript.
A: Done. We grouped the studies into subsections to make it clearer. Now, results on Culex pipiens and Cx. restuans are separately reported. In addition, we have summarized the main results of these studies in the new Table 1. The mechanisms explaining the reported associations are also discussed in the main text.
In the "Conclusions and future directions" section, the authors discuss the correlative and experimental studies, the absence of patterns regarding some studies, and the negative correlation in others. Why is that? I would like to know the author's thoughts on that, and I suggest adding this information to the text.
A: We have extended the manuscript by discussing the potential mechanisms underlying the results obtained by the cited works based on the current knowledge on the field.
The figure 1 needs some work. It is hard to interpret and not well discussed in the text.
A: Following your suggestion, this figure has been removed from the manuscript. Instead, we have included a new table summarizing the main results of the articles included in this review (please, see Table 1).
Lines 63-73: Italicize scientific names.
A: It appears to be a typographical error when converting the submitted Word to pdf. We apologize for this mistake. We have carefully the revised the pdf file before its submission.
Line 97: typo on quinquefasciatus
A: Done (line 152)
Line 189: Which mosquito are the authors discussing regarding Dengue and Chromobacterium? I suggest rephrasing it.
A: The authors of this study refers to Anopheles gambiae. We have extended the section Mosquito microbiota and pathogen transmission, adding general information on interaction between mosquito microbiota and pathogen transmission including this article (lines 139). We therefore remove this sentence from the conclusions to focus the paragraph in WNV.
Round 2
Reviewer 2 Report
Comments and Suggestions for Authors
Considering that the author has made effective improvements to the manuscript, I agree that this review can be accepted for publication, but I still hope that the author will carefully review the entire text to avoid some minor errors. For example, in lines 85-96, none of the species names are italicized.
Author Response
Considering that the author has made effective improvements to the manuscript, I agree that this review can be accepted for publication, but I still hope that the author will carefully review the entire text to avoid some minor errors. For example, in lines 85-96, none of the species names are italicized.
ANSWER: It appears to be a typographical error when converting the submitted Word file to pdf. We apologize for that. We have retyped that paragraph hopefully correcting the error (please see lines 95-106).
Reviewer 3 Report
Comments and Suggestions for Authors
The authors made a significant improvement in the manuscript. I just have one more comment:
Line 245: Based on the new table, you can't discriminate the correlative from experimental studies thus referencing the table here does not make sense. I suggest to rephrase it or deleting the reference to the table.
Author Response
The authors made a significant improvement in the manuscript. I just have one more comment:
Line 245: Based on the new table, you can't discriminate the correlative from experimental studies thus referencing the table here does not make sense. I suggest to rephrase it or deleting the reference to the table.
ANSWER: We provided the information on the type of study in the second column of the Table 1. Additionally, we have modified the text citation of Table 1 (please see line 266).